# Awareness of hepatitis B post-exposure prophylaxis among healthcare providers in Wakiso district, Central Uganda

John Bosco Isunju[1], Solomon Tsebeni Wafula[1], Rawlance Ndejjo[1], Rebecca Nuwematsiko[1], Pamela Bakkabulindi[2], Aisha Nalugya[1], James Muleme[1], Winnie Kansiime Kimara[1], Simon P. S. Kibira[3], Joana Nakiggala[1], Richard K. Mugambe[1], Esther Buregyeya[1], Tonny Ssekamatte[1] *, Rhoda K. Wanyenze[1]

1 Department of Disease Control and Environmental Health, School of Public Health, College of Health Science, Makerere University, Kampala, Uganda, 2 Center of Excellence for Maternal New-Born Child Health Care, School of Public Health, College of Health Science, Makerere University, Kampala, Uganda, 3 Department of Community Health and Behavioural Sciences, School of Public Health, College of Health Science, Makerere University, Kampala, Uganda

* ssekamattet.toca@gmail.com, tssekamatte@musph.ac.ug

**Data Availability Statement:** All supplementary files are available from the plose one database https://doi.org/10.1371/journal.pone.0235470.

## Abstract

### Background

Healthcare providers (HCPs) are at an elevated occupational health risk of hepatitis B virus infections. Post-exposure prophylaxis (PEP) is one of the measures recommended to avert this risk. However, there is limited evidence of HCPs' awareness of hepatitis B PEP. Therefore, this study aimed to establish awareness of hepatitis B PEP among HCPs in Wakiso, a peri-urban district that surrounds Uganda's capital, Kampala.

### Methods

A total of 306 HCPs, selected from 55 healthcare facilities (HCFs) were interviewed using a validated structured questionnaire. The data were collected and entered using the Kobo Collect mobile application. Multivariable binary logistic regression was used to establish the factors associated with awareness of hepatitis B PEP.

### Results

Of the 306 HCPs, 93 (30.4%) had ever heard about hepatitis B PEP and 16 (5.2%) had ever attended training where they were taught about hepatitis B PEP. Only 10.8% were aware of any hepatitis B PEP options, with 19 (6.2%) and 14 (4.6%) mentioning hepatitis B immunoglobulin (HBIG) and hepatitis B vaccine, respectively as PEP options. Individuals working in the maternity department were less likely to be aware of hepatitis B PEP (AOR = 0.10, 95% CI = 0.02–0.53). There was a positive association between working in a healthcare facility in an urban setting and awareness of hepatitis B PEP (AOR = 5.48, 95% CI = 1.42–21.20). Hepatitis B screening and vaccination were not associated with awareness of PEP.

s001 https://doi.org/10.1371/journal.pone.
0235470.s002.

**Funding:** The author(s) received no specific
funding for this work.

**Competing interests:** The authors have declared
that no competing interests exist.

## Conclusions

Only one-tenth of the HCPs were aware of any hepatitis B PEP option. Awareness of hepatitis B PEP is associated with the main department of work and working in a healthcare facility in an urban setting. This study suggests a need to sensitise HCPs, especially those in rural HCFs and maternity wards on hepatitis B PEP. The use of innovative strategies such as e-communication channels, including mobile text messaging might be paramount in bridging the awareness gap.

## Background

Hepatitis B virus infection remains a global health challenge [1]. Hepatitis B is a viral infection transmitted through contact with infected blood or other body fluids such as saliva, menstrual, vaginal, and seminal fluids [1]. Chronic hepatitis B infection remains one of the most serious of viral hepatitis and is often associated with hepatocellular necrosis, inflammation, cirrhosis and hepatocellular carcinoma the major complications [1]. Current evidence indicates that more than 1.5 million new hepatitis B infections are reported annually. In addition, the number of people living with chronic hepatitis B infection increased from 257 million in 2015 to 296 million in 2019 [1, 2]. More than 820,000 hepatitis B-related deaths were reported in 2019, most of which were attributed to cirrhosis and hepatocellular carcinoma [2]. Sub-Saharan Africa (SSA) and Asia bear the greatest burden of chronic hepatitis B, accounting for 68% of hepatitis B infected individuals worldwide [1].

Although the trends in mortality due to human immunodeficiency virus (HIV), tuberculosis and malaria have been decreasing over the years, mortality from viral hepatitis continues to rise, with SSA and Asia registering the highest numbers [3]. Hepatitis B infection is highly endemic in Uganda, with a national prevalence of 10%, and spatial variations across the country ranging from 4% in the southwest, 5% in Kampala and surrounding districts (including Wakiso) to 25% in the northeast region [4]. More than 1,206 hepatitis B-related deaths were reported in Uganda in 2019 [5]. Hepatitis B infection accounts for 80% of liver cancers reported at Uganda's main national referral hospital (Mulago hospital) annually [6]. These outcomes pose a serious economic burden not only to the healthcare system but also at the family level [7–9], and patients are reported to have a low health-related quality of life and catastrophic health expenditure [10–12].

Healthcare providers in SSA are at an elevated occupational risk of hepatitis B infection due to the high prevalence of the disease in the community and the nature of their work [13, 14]. Healthcare providers have an up to four-fold increased risk of acquiring hepatitis B infection compared to the general population [15, 16], due to frequent percutaneous and mucosal exposure to infected blood and bodily fluids [17]. Injection practices worldwide and especially in lower middle–income economies include multiple, available unsafe practices. Unsafe practices but are not limited to prevalent and high-risk practices, include: a) reuse of injection equipment to administer injections to more than one person; b) accidental needlestick injuries in HCPs; c) overuse of injection to health conditions where oral formulations are available; d) unsafe sharps waste management [17–21]. Prevention of hepatitis B infection in healthcare settings includes hand hygiene, safe handling and disposal of sharps and waste, safe cleaning of equipment, testing of donated blood, improved access to safe blood, and training the health personnel [22]. This paper, therefore, provides an opportunity to pass to the healthcare

providers a clear message regarding awareness of hepatitis B of prevention [19, 23]. High costs related to the provision of hepatitis B immunoglobulin during the prevention of mother-to-child transmission further escalate the risk of infection among HCPs in SSA [24].

Despite evidence of the burden of hepatitis B infection, awareness of the disease and uptake of prevention services such as screening and vaccination remain sub-optimal among HCPs [14, 25]. Recent evidence indicates that only three-quarters of HCPs in Wakiso district had ever been screened for hepatitis B infection while more than half were fully vaccinated by 2018 [25]. A high proportion of HCPs had limited knowledge of hepatitis B infection, had a negative attitude and exhibited poor preventive practices [14]. Low knowledge and negative attitudes toward prevention often increase non-adherence to prevention measures such as following standard precautions and PEP, resulting in high rates of hepatitis B infection [14]. The Ugandan Ministry of Health recommended routine vaccination, screening of donor blood and blood products for hepatitis B before transfusion, safe and appropriate use of injections, and adherence to infection prevention and control protocols as measures to reduce hepatitis B infections among HCPs [26]. Additionally, the Uganda blood transfusion service screens all blood from donors following robust quality control measures for transmissible infections such as hepatitis B [27].

The World Health Organization (WHO) recommends vaccination of high-risk groups such as HCPs as a pre-and post-prophylactic measure for hepatitis B prevention [1]. Post-exposure prophylaxis is effective in the prevention of hepatitis B infection and the subsequent development of severe complications if provided appropriately and timely [28–30]. Hepatitis B PEP includes the prevention of perinatal and early childhood hepatitis B infection, persons who inject drugs, men who have sex with men, sex workers and healthcare providers [31, 32]. The prevention strategy involves the provision of a single dose of hepatitis B immunoglobulin (HBIG) to unvaccinated exposed persons within 24 hours of exposure, followed by three doses of hepatitis B vaccine over six months [31]. The administration of HBIG provides primary protection to individuals who are unable to respond to the hepatitis B vaccine in the event of hepatitis B exposure [33].

Owing to the occupational risk of hepatitis B infection among HCPs, the Ugandan Ministry of Health developed national policy guidelines on PEP for hepatitis B, C and HIV [34]. These guidelines provide information on PEP practice, management of exposures and training of healthcare providers regarding the appropriate use of PEP [34]. The ministry particularly recommended the use of PEP during emergency situations such as the prevention of mother-to-child transmission, and occupational exposure for HCPs [26, 30]. Despite these guidelines, poor and/or incorrect PEP practices are still prevalent in Uganda, similar to many other healthcare settings [35, 36]. There is evidence that some exposures to infectious blood or body fluids among HCPs often go unnoticed and, even if exposures are recognised, HCPs often do not seek PEP [35, 36]. Despite this evidence, little is known about HCPs' awareness of hepatitis B PEP in low-income settings such as Uganda. Therefore, this study established awareness of hepatitis B PEP among HCPs in Wakiso district, Central Uganda. Our findings can be used as a basis for creating awareness and consequently leading to the utilisation of PEP and procurement of PEP. Furthermore, our findings can also be used to inform curriculum reviews for HCPs' training programmes, and content of continuous medical education sessions.

## Methods

### Study setting

This study was conducted among HCPs in Wakiso District, Central Uganda. According to the 2014 population census, Wakiso District is a predominantly rural area, with a population of

approximately 2,007,700 inhabitants. The district has seven health sub-districts with 533 HCFs (10 hospitals, 15 Health centres (HCs) IVs, 156 HCIIIs, and 232 HCIIs) [37]. Healthcare facilities in Uganda start at level 1, which is designated as HC I, to HC II, III, IV, general hospitals, regional referral hospitals and national referral hospitals. Wakiso District has a regional referral hospital (Entebbe regional referral hospital). The catchment population and services offered at the various levels are indicated in Table 1.

## Study design and sample size estimation

This cross-sectional study was conducted in July 2018 and employed quantitative data collection methods. The sample size was calculated using the Kish Leslie sample size formula for cross-sectional studies [39]. The assumptions for the sample size calculation were a prevalence (p) of adequate knowledge of hepatitis B PEP of 12.1% [19], a 95% level of confidence, an error rate (d) of 0.05 and a Z score of 1.96 corresponding to the two 95% confidence interval (CI) and a design effect of 2.0. This yielded a final sample size of 325.

## Sampling procedures and data collection

The detailed sampling procedure was reported in our previous studies [14, 25]. Briefly, we purposively considered 6 general hospitals and 16 HC IVs since these serve a large proportion of the population and also offer high-risk medical interventions such as caesarean deliveries and blood transfusion. These procedures expose HCPs to an elevated risk of hepatitis B infection. General hospitals and HC IVs were either private for profit, private not for profit or public or public (government) HCFs. We randomly selected 33 HC IIIs from the district HCF inventory.

 The sample size was distributed proportionate to the number of HCPs employed at the selected HCFs and their availability during the survey period. Before conducting individual structured interviews, a list of all HCPs was obtained from the HCF administrator or in charge to form a sampling frame for each HCF. Simple random sampling was then used to select HCPs at each HCF to respond to the standardised English questionnaire developed by experts guided by the reviewed literature. The tool was first pretested among HCPs in HCFs in Mukono district. The selected HCFs considered for pretesting had characteristics similar to

**Table 1. Catchment population and services offered across the different healthcare facility levels in Uganda.**

| No | Level | Catchment population | Services provided |
|----|-------|---------------------|-------------------|
| 1 | Clinic/Health centre I | Undefined | Community-based preventive and promotive health services such as village health teams or similar status. |
| 2 | Health centre II | 5,000 | Preventive, promotive and outpatient curative health services, outreach care, and emergency |
| 3 | Health centre III | 20,000 | Preventive, promotive, outpatient curative, maternity, inpatient health services and laboratory services |
| 4 | Health centre IV | 100,000 | Preventive, promotive, outpatient curative, maternity, inpatient health services, emergency surgery and blood transfusion and laboratory services |
| 5 | General hospital | 500,000 | In addition to services offered at healthcare centre IV, other general services are provided. These facilities also provide in service training, consultation and research |
| 6 | Referral hospital | 1,000,000 | In addition to services offered at the general hospital, these offer a package of specialised services and training |
| 7 | Regional referral hospital | 2,000,000 | In addition to services offered at the general hospital, these offer specialist services such as psychiatry, ear, nose and throat, ophthalmology, dentistry, intensive care, radiology, pathology, higher level surgical. |
| 8 | National referral hospital | 10,000,000 | These provide comprehensive specialist services. In addition, they are involved in teaching and research. |

Source: National Health Facility Master List 2018 [38].

those of the study area. Questionnaires were used to obtain detailed information on socio-demographics, screening and vaccination status and knowledge of prophylactic management of hepatitis B infection. The questionnaires were administered by experienced and trained research assistants upon obtaining written informed consent from the participants.

## Study variables

The dependent variable was awareness of hepatitis B PEP options. A participant was considered aware of the hepatitis B PEP options if they mentioned either HBIG or hepatitis B vaccine or both. Other parameters related to knowledge of PEP; ever hearing about PEP; source of information about PEP; history of attending training on PEP among others. The independent variables included sociodemographic factors such as age of HCP, duration of work experience, the highest level of education, area of medical specialisation (cadre), department of work, history of injury, position at the HCF, years of training and institution of training. HCF was considered rural if it was located in a sub-county and urban if it was located in a town council or municipality. Healthcare providers were classified as "married" if they were legally married or cohabiting and "not married" if they were not in any union.

## Data management and statistical analyses

Data were collected and entered using the KoboCollect mobile application, and synchronised daily onto the server. Mobile data collection using KoboCollect permits real-time data capture and entry and minimises errors throughout the data management process [25, 40]. The data entry screens were designed with skips and restrictions to ensure quality and completeness. To ensure that the data were secure, only the principal investigators had the security key for the KoboCollect server hosted at https://www.kobotoolbox.org/ where the data were sent after synchronisation. Data were then exported for analysis in Stata 16.0 statistical software (Statacorp, College station, Texas, USA). Data were then summarised as frequencies, percentages, means and standard deviations where applicable. Since the outcome variable (awareness of PEP options for hepatitis B) was dichotomous and had a low prevalence ($<10\%$), we performed a multivariable logistic regression to assess the dependence of awareness of PEP for hepatitis B on sociodemographic and individual factors. Initially, simpler regression models consisting of the outcome and one predictor at a time were run to produce unadjusted odds ratios. Variables with *p values* less than 0.25 in the bivariable models and those with literature backup evidence were added into the multivariable model while adjusting for age and sex. Statistical significance was set at $P \leq 0.05$. Both unadjusted and adjusted odds ratios (AORs) and their corresponding 95% confidence intervals are reported in this study.

## Quality assurance and quality control

Data collectors were recruited from our well-established network of research assistants who had participated in previous successful research projects. All research assistants underwent a 3-day training on the research protocol and ethical issues surrounding the study to ensure quality data collection. The data collection tools were pre-tested among the 10 HCPs in Kampala district. Kampala was purposively selected because it shares similar characteristics with Wakiso district, such as being highly populated. Pre-testing of the tools enabled the team to correct any errors in the tools, minimise ambiguity, improve validity and enabled the RAs to familiarise themselves with the data collection tools.

### Ethical considerations

Ethical approval for the study was obtained from the Makerere University School of Public Research and Ethics Committee. Administrative clearance was sought from the Wakiso district Local government and management of the participating HCFs. Written informed consent was obtained from the study respondents before any interviews were conducted. All informed consent discussions were conducted in English since all the HCP were literate.

## Results

### Demographic characteristics of respondents

A total of 306 HCPs completed the survey, representing a response rate of 94.1%. Of these, 206 (67.3%) were females, 207 (60.8%) were aged between 20 and 30 years with a median age of 27 years (IQR 24, 33). A large proportion, 204 (66.7%) worked at HCFs in urban settings and had a medium working experience of 4 years (IQR 2, 5) (Table 2). Only 16 (29.1%) of the HCFs had received hepatitis B vaccine doses in the last 12 months.

### Awareness of PEP for hepatitis B infection

A total of 93 (30.4%, 95CI: 25.5% - 35.8%) HCPs had heard about PEP for hepatitis B infection, with the main source of information being HCFs, 59 (63.4%) and media 12 (12.9%). Only 109 (35.6%, 95%CI 30.4% - 41.1%) had ever heard about HBIG. Most respondents 292 (95.4%) considered themselves at risk of acquiring hepatitis B infection and 49 (16.0%) had needle pricks in the last 12 months. Only 16 (5.2%) HCPs had ever received training on PEP for

**Table 2. Socio-demographic characteristics of respondents.**

| Variable | Category | n | Percentage (%) |
|---|---|---|---|
| Sex | Female | 206 | 67.3 |
| | Male | 100 | 32.7 |
| Age in years | 20–30 | 207 | 67.7 |
| | 31–40 | 70 | 22.9 |
| | ≥40 | 29 | 9.5 |
| | Median (IQR) | 306 | 27 (24, 33) |
| Marital status | Married | 128 | 41.8 |
| | Not married | 178 | 58.2 |
| Years of experience as HCP | ≤3 | 146 | 47.7 |
| | 4–6 | 74 | 24.2 |
| | 7–10 | 46 | 15.0 |
| | >10 | 40 | 13.1 |
| | Median (IQR) | 306 | 4 (2, 7) |
| Level of HCF | Health centre III | 133 | 43.5 |
| | Health centre IV | 120 | 39.2 |
| | Hospital | 53 | 17.2 |
| Ownership of HCF | Private for profit | 136 | 44.4 |
| | Private not for profit | 30 | 9.8 |
| | Public | 140 | 45.7 |
| Location of HCF where HCP works | Rural | 102 | 33.3 |
| | Urban | 204 | 66.7 |

HCP: Healthcare provider; HCF: Healthcare facility; IQR: Interquartile range

hepatitis B infection. Moreover, about 33 (10.8%, 95%CI 7.8% - 14.8%) of HCPs were aware of the PEP options for hepatitis B infection. Of these, 19 (6.1%, 95%CI 3.8% - 9.5%) mentioned HBIG and 14 (4.6%, 95%CI 2.5% - 7.6%) mentioned hepatitis B infection vaccine (Table 3).

## Factors associated with awareness of hepatitis B infection post-exposure prophylaxis

At bivariable regression, working in the maternity ward, in urban health facilities, being vaccinated for hepatitis B infection and having knowledge of hepatitis B infection were associated with knowledge of PEP options for hepatitis B infection. After adjusting for age and gender during multivariable modelling, HCPs working in the maternity ward were 89% less likely to be aware of any PEP options for hepatitis B infection (AOR = 0.11, 95%CI = 0.02–0.57). The odds of being knowledgeable about PEP options was 5.5 times among HCPs working in urban settings when compared with those in rural health facilities (AOR = 5.56, 95%CI = 1.47–20.99) (Table 4).

**Table 3. Awareness of hepatitis B infection post-exposure prophylaxis among healthcare providers in Wakiso district, Uganda.**

| Variable | Category | n | Percentage (%) |
|---|---|---|---|
| Ever heard about hepatitis B infection PEP (N = 306) | Yes | 93 | 30.4 |
| | No | 213 | 69.6 |
| Source of information on PEP (n = 93) | HCF | 59 | 63.4 |
| | Media | 12 | 12.9 |
| | Workshops/Outreaches | 14 | 15.1 |
| | Others (including training school) | 8 | 8.6 |
| Ever received a training on PEP (n = 306) | Yes | 16 | 5.2 |
| | No | 290 | 94.8 |
| Ever heard of HBIG | Yes | 109 | 35.6 |
| | No | 197 | 64.4 |
| HBIG is administered intravenous or intramuscularly (N = 109) | Yes | 65 | 59.6 |
| | No | 44 | 40.4 |
| HBIG provides short term protection against hepatitis B infection (N = 109) | Yes | 42 | 38.5 |
| | No | 67 | 61.5 |
| Aware of any hepatitis B infection PEP options | No | 273 | 89.2 |
| | Yes | 33 | 10.8 |
| HBIG is used for PEP | Yes | 19 | 6.2 |
| | No | 287 | 93.8 |
| Hepatitis B infection vaccine can be used for PEP | Yes | 14 | 4.6 |
| | No | 292 | 95.4 |
| Considered themselves at risk | Yes | 292 | 95.4 |
| | No | 14 | 4.6 |
| Had a needle prick in the last 12 months | Yes | 49 | 16.0 |
| | No | 257 | 84.0 |
| Hepatitis B infection is treatable | Yes | 270 | 88.2 |
| | No | 36 | 11.8 |

PEP: post-exposure prophylaxis; HCP: Healthcare provider; HCF: Healthcare facility; HBIG: Hepatitis B immunoglobulin

Table 4. Factors associated with awareness of hepatitis B infection post exposure prophylaxis among healthcare providers in Wakiso district, Uganda.

| Variable | Aware of hepatitis B infection PEP | | Crude OR (95% CI) | p-value | Adjusted OR (95% CI) | p-value |
|---|---|---|---|---|---|---|
| | Yes n (%) | No n (%) | | | | |
| **Sociodemographic characteristics** | | | | | | |
| **Sex** | | | | | | |
| Female | 18 (8.7) | 188 (91.3) | 1 | | 1 | |
| Male | 15 (15.0) | 85 (85.0) | 1.84 (0.87–3.83) | 0.101 | 1.05 (0.45–2.49) | 0.900 |
| Age of respondent (years) | | | | | | |
| ≤30 | 22 (10.6) | 185 (89.4) | 1 | | | |
| 31–40 | 9 (12.9) | 61 (87.1) | 1.24 (0.54–2.84) | 0.610 | 1.74 (0.71–4.28) | 0.228 |
| 41 and above | 2 (6.9) | 27 (93.1) | 0.62 (0.14–2.80) | 0.537 | 0.86 (0.17–4.23) | 0.848 |
| **Department of work** | | | | | | |
| In patient clinic | 10 (21.3) | 37 (78.7) | 1 | | 1 | |
| Maternity ward | 2 (2.4) | 82 (97.6) | 0.09 (0.02–0.43) | **0.003** | 0.11 (0.02–0.57) | **0.009** |
| Outpatient clinic | 21 (12.0) | 154 (88.0) | 0.50 (0.22–1.16) | 0.108 | 0.56 (0.22–1.43) | 0.227 |
| **Cadre** | | | | | | |
| Clinical officer /general practitioners | 9 (11.0) | 73 (89.0) | 1 | | | |
| Nurses/midwives | 10 (9.2) | 99 (90.8) | 0.82 (0.32–2.11) | 0.681 | | |
| Anaesthetist | 2 (6.7) | 28 (93.3) | 0.58 (0.12–2.84) | 0.502 | | |
| Lab personnel and other cadres* | 12 (14.1) | 73 (85.9) | 1.33 (0.52–3.35) | 0.541 | | |
| **Years of experience as HCP** | | | | | | |
| ≤3 | 22 (11.1) | 176 (88.9) | 1 | | | |
| ≥4–6 | 11 (10.2) | 97 (89.8) | 1.79 (0.76–4.21) | 0.285 | | |
| 7–10 | | | 0.71 (0.19–2.62) | 0.612 | | |
| 11 and above | | | 1.81 (0.64–5.09) | 0.265 | | |
| **Healthcare level** | | | | | | |
| Health centre II-III | 10 (7.5) | 123 (92.5) | 1 | | 1 | |
| Health centre IV | 17 (14.2) | 103 (85.8) | 2.03 (0.89–4.63) | 0.092 | 1.10 (0.41–2.93) | 0.9855 |
| Hospital | 6 (11.3) | 47 (88.7) | 1.57 (0.54–4.56) | 0.407 | 1.29 (0.38–4.33) | 0.683 |
| **Ownership of facility** | | | | | | |
| Private | 18 (13.2) | 118 (86.8) | 1 | | 1 | |
| PNFP | 04 (13.3) | 26 (86.7) | 1.01 (0.32–3.23) | 0.989 | 0.68 (0.17–2.66) | 0.7575 |
| Public | 11 (7.9) | 129 (92.1) | 0.56 (0.25–1.23) | 0.149 | 0.73 (0.31–1.74) | 0.475 |
| **Location** | | | | | | |
| Rural | 3 (2.9) | 99 (97.1) | 1 | | 1 | |
| Urban | 30 (14.7) | 174 (85.3) | 5.69 (1.69–19.1) | **0.005** | 5.56 (1.47–20.9) | **0.011** |
| **Marital status** | | | | | | |
| Married | 15 (11.7) | 113 (88.3) | 1 | | | |
| Not married | 18 (10.1) | 160 (89.9) | 0.85(0.41–1.75) | 0.655 | | |
| **Knowledge /perception** | | | | | | |
| **Know hepatitis B infection is treatable** | | | | | | |
| No | 4 (11.1) | 32 (88.9) | | | | |
| Yes | 29 (10.7) | 241 (89.3) | 0.96 (0.32–2.92) | 0.946 | | |
| **Ever trained on PEP for hepatitis B infection** | | | | | | |
| No | 32 (11.0) | 258 (89.0) | 1 | | | |
| Yes | 1 (6.3) | 15 (93.7) | 0.54 (0.07–4.21) | 0.554 | | |
| **Belief that their job puts them at high risk** | | | | | | |
| No | 1(20.0) | 4 (80.0) | 1 | | | |

(*Continued*)

**Table 4.** (Continued)

| Variable | Aware of hepatitis B infection PEP | | Crude OR (95% CI) | p-value | Adjusted OR (95% CI) | p-value |
|---|---|---|---|---|---|---|
| | Yes n (%) | No n (%) | | | | |
| Yes | 32 (10.6) | 269 (89.4) | 0.48 (0.05–4.39) | 0.512 | | |
| **Considered themselves to be at risk of hepatitis B infection** | | | | | | |
| No | 2 (14.3) | 12 (85.7) | 1 | | | |
| Yes | 31 (10.6) | 261 (89.4) | 0.71 (0.15–3.33) | 0.667 | | |
| **Ever screened for hepatitis B infection** | | | | | | |
| No | 3 (4.0) | 73 (96.0) | 1 | | 1 | |
| Yes | 30 (13.0) | 200 (87.0) | 3.65 (1.08–12.32) | 0.037 | 1.58 (0.64–3.95) | 0.319 |

CI: Confidence interval; PNFP: Private Not for Profit; OR: Odds ratio; PEP: post-exposure prophylaxis; HCP: Healthcare provider; Other cadres* include; dental officers, pharmacists, opticians, counsellors.

## Discussion

This study determined the awareness of hepatitis B infection post-exposure prophylaxis among healthcare providers in Wakiso district–a peri-urban district that encircles Uganda's capital Kampala. The study found low levels of awareness of hepatitis B infection PEP among HCPs in Wakiso. Less than one-third of the HCPs knew that hepatitis B infection had PEP. The low awareness of hepatitis B PEP may have resulted from a lack of training on the prevention of hepatitis B infection. The Uganda National Policy guidelines on PEP recommend training HCPs as a key strategy for ensuring proper management practices upon exposure to hepatitis B infection [34]. Despite this recommendation, only 5.2% of the HCPs in this study had ever received a training where they were sensitised about hepatitis B PEP. The limited training opportunities reduce the chances for sharing information in the event that some HCPs are unaware of the risk of hepatitis B infection. In such scenarios, HCPs may be inclined to only obtaining PEP for HIV, yet they are equally vulnerable to hepatitis B infection. The findings in our study are not different from those in the Tamale metropolis, Ghana, where only 12.1% of HCPs were aware of HBIG and hepatitis B vaccine as PEP options for hepatitis B [19].

The administration of HBIG provides primary protection after exposure to hepatitis B among individuals who do not respond to hepatitis B vaccination or among unvaccinated exposed individuals [41]. However, this was less known by the HCPs in our study. More than a third of HCPs were unaware of HBIG; with a slight majority wrongfully reporting that it provides long term protection against hepatitis B infections. In addition, only 4.6% of the HCPs were aware that the hepatitis B vaccine could be used as a hepatitis B PEP. Awareness about HBIG as a PEP option was slightly higher (6.2%) in this study than in a study in Ghana (2.8%), while awareness about hepatitis B vaccine as a PEP option was slightly lower (4.6%) in this study compared to the Ghana study (9.3%) [19]. These findings, therefore, signal the need to sensitise HCPs on the different hepatitis B PEP options which are important measures for reducing the risk of hepatitis B infection in the event of exposure.

In our study, HCPs who mainly worked in the maternity ward were less likely to be aware of the hepatitis B PEP options compared to their counterparts in the inpatient department. This is of concern given that the risk of exposure to bloodborne infections can be high in either department. Maternity wards in Ugandan HCFs are characterised by a heavy workload [42], which could prevent some HCPs in the department from attending capacity building

programs aimed at improving their awareness of hepatitis B prevention strategies. Therefore, the use of innovative strategies such as e-communication channels, including mobile text messaging might be paramount in bridging the awareness gap.

Healthcare providers in urban HCPs were more likely to be aware of hepatitis B PEP compared to their counterparts in the rural HCFs. The high awareness of hepatitis B PEP reported among HCPs in urban HCFs could be attributed to the increased opportunities to access information in urban settings. Urban areas usually have better access to communication channels such as the internet, training opportunities, and researchers and policymakers who may act as reliable sources of information on hepatitis B PEP. Lien, Chuc [43] also contends that better access to information is associated with a higher prevalence of awareness of hepatitis B PEP among HCPs in urban settings.

We expected that HCPs who had ever been screened or vaccinated would be more aware of hepatitis B PEP compared to those who had never been screened or vaccinated. However, we did not find a significant association between having ever been screened or vaccinated against hepatitis B infection and awareness of hepatitis B PEP. Our findings imply that vaccination and screening services have not been effectively used to relay information on hepatitis B PEP to HCP. It should be noted that hepatitis B screening and vaccination provide an opportunity for the dissemination of information related to the prevention and management of the disease to HCPs [44, 45]. Therefore, there is a need for those involved in the provision of screening and vaccination services to provide adequate information on all the prevention options, including the use of PEP.

The low level of awareness of hepatitis B PEP reported in the current study is alarming, considering the high risk of hepatitis B infection that characterises healthcare settings. More than a tenth of the HCPs in our study reported needlestick injuries in the last 12 months, which is indicative of the high risk of bloodborne infections. Without behavioural change, and access to PEP, a significant proportion of the exposures could turn into hepatitis B infections among HCPS [35, 46–48], hence ultimately impact the health of HCPs and health service delivery.

## Strengths and limitations

This is one of the few studies that has so far established knowledge and awareness of hepatitis B PEP. Compared to the few studies conducted, it used a relatively large sample size which makes our findings more generalisable. Our study included HCPs in private HCFs. These are rarely studied yet they immensely contribute to service delivery. We didn't attain the required sample size which may have affected the statistical power of the study. However, we had a high response rate which makes our findings reliable.

## Conclusions

Only a tenth of the HCPs in Wakiso district was aware of any hepatitis B PEP option, yet several HCPs had ever suffered a needlestick injury which could elevate their risk of blood-borne infections, including hepatitis B. Healthcare provider's awareness of hepatitis B PEP was associated with the main department of work and location of the healthcare facility. On the contrary, screening and vaccination were not associated with HCP awareness of hepatitis B PEP. Our findings suggest the need to use screening and vaccination opportunities to sensitise HCPs on the need and availability of hepatitis B PEP options for hepatitis B infection, especially those working in rural HCFs and maternity wards. The hepatitis B PEP knowledge gaps identified in the current study should be used as a basis for informing the curriculum for health training programmes and the content of continuous medical education for HCPs. The

use of innovative strategies such as e-communication channels, including mobile text messaging might be paramount in bridging the awareness gap.

## Supporting information

**S1 Appendix. Health care providers' hepatitis b vaccination status and their level of knowledge attitude and practice towards prophylactic management of hbv: A crossectional survey in Wakiso district.**
(DOCX)

**S1 Dataset.**
(XLS)

**S1 File.**
(XLS)

## Acknowledgments

The authors would like to thank the Wakiso district health office for granting permission to conduct the study and all the healthcare providers for their voluntary participation in the data collection process. We also remain indebted to our research assistants (Atukunda Jadrine, Nabukenya Dorothy, Wagaba Brenda, Nakintu Daffine and Nalukenge Dorah) whose due diligence was key in the successful completion of the study.

## Author Contributions

**Conceptualization:** Rawlance Ndejjo, Pamela Bakkabulindi, Aisha Nalugya, James Muleme, Winnie Kansiime Kimara, Simon P. S. Kibira, Joana Nakiggala, Richard K. Mugambe, Esther Buregyeya, Tonny Ssekamatte, Rhoda K. Wanyenze.

**Formal analysis:** Solomon Tsebeni Wafula, Rawlance Ndejjo, Rebecca Nuwematsiko, Pamela Bakkabulindi, Aisha Nalugya, James Muleme, Winnie Kansiime Kimara, Simon P. S. Kibira, Joana Nakiggala, Richard K. Mugambe, Esther Buregyeya, Tonny Ssekamatte, Rhoda K. Wanyenze.

**Methodology:** John Bosco Isunju, Solomon Tsebeni Wafula, Rebecca Nuwematsiko, Simon P. S. Kibira, Richard K. Mugambe, Tonny Ssekamatte, Rhoda K. Wanyenze.

**Project administration:** John Bosco Isunju, Joana Nakiggala, Richard K. Mugambe, Tonny Ssekamatte, Rhoda K. Wanyenze.

**Resources:** John Bosco Isunju, Richard K. Mugambe.

**Supervision:** John Bosco Isunju, Aisha Nalugya, Tonny Ssekamatte.

**Writing – original draft:** John Bosco Isunju, Solomon Tsebeni Wafula, Rawlance Ndejjo, Rebecca Nuwematsiko, Pamela Bakkabulindi, Aisha Nalugya, James Muleme, Winnie Kansiime Kimara, Simon P. S. Kibira, Joana Nakiggala, Richard K. Mugambe, Esther Buregyeya, Tonny Ssekamatte, Rhoda K. Wanyenze.

**Writing – review & editing:** John Bosco Isunju, Solomon Tsebeni Wafula, Rawlance Ndejjo, Rebecca Nuwematsiko, Pamela Bakkabulindi, Aisha Nalugya, James Muleme, Winnie Kansiime Kimara, Simon P. S. Kibira, Joana Nakiggala, Richard K. Mugambe, Esther Buregyeya, Tonny Ssekamatte, Rhoda K. Wanyenze.

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
