## [Decision Letter · Decision Letter 0]

6 Oct 2021

PONE-D-21-16912Awareness of Hepatitis B Post-Exposure Prophylaxis among Healthcare providers in Wakiso district, Central Uganda.PLOS ONE

Dear Dr. Ssekamatte,

Thank you for submitting your manuscript to PLOS ONE. After careful consideration, we feel that it has merit but does not fully meet PLOS ONE’s publication criteria as it currently stands. Therefore, we invite you to submit a revised version of the manuscript that addresses the points raised during the review process.

We look forward to receiving your revised manuscript.

Kind regards,

Orvalho Augusto, MD, MPH

Academic Editor

PLOS ONE

Journal Requirements:

2. Please ensure you have discussed the limitations of this study within the Discussion section, including any potential bias introduced during data collection.

Additional Editor Comments (if provided):

This is an important report on an assessment of awareness of Hepatitis B Post-Exposure Prophylaxis (the authors abbreviate PEP) prevalence among health care providers (HCP) somewhere in Uganda. The authors offer a good background on the burden of Hepatitis B and a good justification for carrying out such a study. However, the analysis is full of some grave shortcomings as both reviewers point out.

Major:

1. Lack of background on policy or actions for Uganda government in Hepatitis B prevention in health workers. This is important contextual information.

2. Appropriateness of the statistical analysis for the sample design - The design of the sample calculation and procedures suggests being of a cluster sampling:

a. Although there is no indication as to how the cluster sizes were determined i.e how did the investigators decided how many individuals should be chosen in each health facility?

b. The statistical analysis ignores the fact this is a cluster sampling. Are the authors aware of this?

c. A few variables should not be considered as predictors. For example, “Hepatitis B can be vaccinated against” and “vaccinated for hepatitis B” are part of the outcome, right? They should not be in table 3.

3. Variable selection for adjustment and parametrisation

a. in the “data management and statistical analyses” sub-section, it is written that age and sex were kept for all adjusted models. It is strange that the multivariable model doesn’t contain age coefficients.

b. Why age and experience time is dichotomized here? Did you assess any non-linearity for such a decision? Even for descriptive purposes in table 1, it does not help. Please reconsider adding more categories for table 1 for these two (or at least add quartiles) and for adjustment (table 3) add age as continuous or age with more categories.

Minor

1. Please enumerate the pages and put line numbers. It is very hard to reference corrections without that.

2. Abstract in the results please add the number of health facilities from where HCP were selected.

3. In the study setting, please add the year for the population.

4. In the study setting, the description of the health care levels causes doubts. For level IV, the current description says that in addition to what lower levels do at this level there are consultations and research. I believe this is incorrect. All health facilities would have at least some form of outpatient consultations, and research, cannot be restricted to higher-level health facilities as documented in many peer-reviewed manuscripts from Uganda.

5. In the “Data management and statistical analyses”

a. please put citation to the KobCollect.

b. Somewhere in “... with low prevalence (<10%). We performed a ...”, there should be a comma replacing the period.

6. Results:

a. Table 1 as explained above we need more categories for age and time experience or consider adding quantiles for descriptive purposes

b. Why the prevalence of the outcome and its components (prevalence of HBIG awareness, prevalence of vaccine awareness and of the combined) does not have a confidence interval? Remember to account for the complex nature of this sample.

c. Table 3 for the models. See the above comments.

7. Discussion: why no limitation discussion here?

Reviewers' comments:

Reviewer's Responses to Questions

**Comments to the Author**

1. Is the manuscript technically sound, and do the data support the conclusions?

Reviewer #1: Partly

Reviewer #2: Partly

2. Has the statistical analysis been performed appropriately and rigorously? 

Reviewer #1: I Don't Know

Reviewer #2: Yes

3. Have the authors made all data underlying the findings in their manuscript fully available?

Reviewer #1: Yes

Reviewer #2: Yes

4. Is the manuscript presented in an intelligible fashion and written in standard English?

Reviewer #1: Yes

Reviewer #2: Yes

5. Review Comments to the Author

Reviewer #1: HBV has long been recognized as an occupational risk for HCP. HCP do not recognize all exposures to potentially infectious blood or body fluids and, even if exposures are recognize, often do not seek post-exposure prophylactic management. Vaccines to prevent HBV are worldwide available and were recommended for HCP since 1982. Acute and chronic HBV infections are rare among HCP who respond to HepB vaccination, but HCP who do not respond to vaccination are thought to remain susceptible. Postvaccination serologic testing for anti-HBs for HCP at risk for needle stick exposures is recommended 1-2 months after completion of the HepB vaccine series.

The study “Awareness of hepatitis B post-exposure prophylaxis among healthcare providers in Wakiso district, Central Uganda” established awareness of hepatitis B PEP among HCPs in one district of Uganda.

The manuscript is technically sound, however to understand the results and the conclusions of the study it will be useful if the authors provide data on HepB policy to prevent HBV in Uganda in general, and in the district in particular:

a) Coverage of hepatitis B virus vaccine

b) Coverage of prevention of mother-to-child transmission of hepatitis virus

c) Blood donations screened in a quality-assured manner

d) Infections administered with safety-engineered devices

Hepatitis protection among HCP is not PEP alone, and a number of strategies include immunization against HepB, universal precautions, control measures, education, and reporting and follow-up of exposure.

Programmes to support Infection, Prevention and Control are particularly important in low-income countries, where health care delivery and medical hygiene standards may be improved.

Infection, Prevention and Control Programmes is the key to fight against infectious diseases in general, and against HepB in particular, and PEP is one of the components of the strategy.

Finally, is the healthcare facilities where interviews have occurred, did serologic tests for HpB, and HBIG and hepatitis B vaccine are available?

The availability of these data is useful to understand better in which context this study was conducted, and the reasons for the low prevalence of awareness of HepB PEP among HCP in Wakiso district, Central Uganda.

Reviewer #2: Is the manuscript technically sound: Yes, although the manuscript would benefit from an English review and editing services to improve the overall flow. In the introduction, it would be important to provide data on the approximate mortality from HBV in Uganda in order to give the reader a better perspective of the burden of the disease and to further justify this study. The authors should consider providing information on the availability of HBIG and Hepatitis Vaccination at health facilities in Uganda; limited access to PEP may also contribute to the little knowledge.

The authors should provide a stronger justification for this study. In the introduction, the authors state "These findings can be used to inform policy and practice related to the prevention of HBV infection among HCPs." There are other reasons why it would be important to study the knowledge on HBV prophylaxis and it would be helpful for the authors to elaborate that in this paper to further convince the reader of the importance of this study.

Methods: The authors note that "Variables with p values less than 0.25 in the bivariable models and those with literature backup evidence were added into the multivariable model while adjusting for age and sex". What informed the choice of p value of 0.25 as a threshold for including variables in the multivariable analysis?

From the power and sample size calculation, the authors calculated a desired sample size of 325. However, data was collected from 306 research participants. The authors have not discussed the potential impact of data collection from a smaller population size than initially calculated.

The authors should be clearer on the definition of the dependable outcome. They have provided a brief definition of what "knowledge on HBV PEP". This definition however needs to be elaborated.

Discussion: Should be expanded to include a discussion on why the various social and demographic factors were selected as independent variables, and the significance of the relationship, significant or not, to the dependant variable.

Recommendations: The authors state that "Our findings suggest the need to use screening and vaccination

pportunities to sensitise HCPs on the PEP options for HBV infection.". There could be other recommendations from this study. The primary recommendation should be to educate health workers on the need and on the availability of HBV PEP, given that 90% of them are not aware. The recommendations could be strengthened to align with the findings from the study.

6. PLOS authors have the option to publish the peer review history of their article (what does this mean?). If published, this will include your full peer review and any attached files.

Reviewer #1: No

Reviewer #2: **Yes: **Griffins Manguro

---

## [Author Response · Author response to Decision Letter 0]

24 Feb 2022

Response to comments

Thank you, reviewers, for providing feedback that has significantly improved the manuscript. Below is the response to comments raised during the review process

Reviewer 1

Major comments

Comment: Lack of background on policy or actions for Uganda government in Hepatitis B prevention in health workers. This is important contextual information.

Response: Thank you. Background information on hepatitis B prevention has been provided. Page 4 lines 108-115

Comment: Appropriateness of the statistical analysis for the sample design - The design of the sample calculation and procedures suggests being of a cluster sampling:

a. Although there is no indication as to how the cluster sizes were determined i.e. how did the investigators decided how many individuals should be chosen in each health facility?

Response: We didn’t do cluster sampling. The design effect was considered because of the fact that we would over sample urban facilities and again, we believed facilities in urban settings would be different from those in rural areas.

Comment: b. The statistical analysis ignores the fact this is a cluster sampling. Are the authors aware of this?

Response: We didn’t consider cluster sampling (as is done in DHS data) and therefore did not need to incorporate survey package in analysis.

Comment: c. A few variables should not be considered as predictors. For example, “Hepatitis B can be vaccinated against” and “vaccinated for hepatitis B” are part of the outcome, right? They should not be in table 3.

Response: Thanks for the pointing this out, we have removed those variables from the model. Page 12

Comment: Variable selection for adjustment and parametrization

a. In the “data management and statistical analyses” sub-section, it is written that age and sex were kept for all adjusted models. It is strange that the multivariable model doesn’t contain age coefficients.

Response: Thanks for this observation, it was an omission, Age was originally adjusted for but results were not written. We have now presented the multivariate results adjusted for age and gender. Page 12

Comment: b. Why age and experience time is dichotomized here? Did you assess any non-linearity for such a decision? Even for descriptive purposes in table 1, it does not help. Please reconsider adding more categories for table 1 for these two (or at least add quartiles) and for adjustment (table 3) add age as continuous or age with more categories.

Response: There was no significant difference in the fit of the model regardless of whether age was continuous or categorical. For the model, we have maintained categorical values. We thought there was no relevance in assessing linearity/non-linearity for age and experience because our outcome is binary /not continuous (hence linear regression was not performed). Pages 12-13 and Page 9

Minor comments

Comment: Please enumerate the pages and put line numbers. It is very hard to reference corrections without that.

Response: We have included page numbers and line numbers in the new submission.

Comment: Abstract in the results please add the number of health facilities from where HCP were selected.

Response: Thank you, 55 health facilities were indicated in the initial submission. Page 2 line 47

Comment: In the study setting, please add the year for the population.

Response: Thanks for the comment, we have now added the year. Page 5 line 129

Comment: In the study setting, the description of the health care levels causes doubts. For level IV, the current description says that in addition to what lower levels do at this level there are consultations and research. I believe this is incorrect. All health facilities would have at least some form of outpatient consultations, and research, cannot be restricted to higher-level health facilities as documented in many peer-reviewed manuscripts from Uganda.

Response: A table indicating the services offered across the various levels of healthcare facilities has been added for more clarity. Page 5 lines 137-138. The consultations referred to in this manuscript is seeking expert advice from a consultant physician/ specialist in a specific area e.g. Gynaecology. The research referred to in this paper refers to a healthcare facility being able to undertake research and not to be used as a research object (or being studied). Some of these facilities have research and ethics committees or scientific committees. That is why it is stipulated that they engage in research.

Comment: In the “Data management and statistical analyses” a. please put citation to the KoboCollect

Response: Thank you. Citation has been added. Page 7 line 178

Comment: b. Somewhere in “... with low prevalence (<10%). We performed a ...”, there should be a comma replacing the period.

Response: Thank you. We have made this change. Page 7 line 186

Comment: Results: a. Table 1 as explained above we need more categories for age and time experience or consider adding quantiles for descriptive purposes

Response: We have provided three categories for age and median age (IQR). We have also put four levels for experience as suggested. Page 9 lines 219-220

Comment: b. Why the prevalence of the outcome and its components (prevalence of HBIG awareness, prevalence of vaccine awareness and of the combined) does not have a confidence interval?

Response: We initially just didn’t provide it but we have now provided confidence intervals. We don’t think it is relevant to provide confidence intervals for very variable in the table. Hence, we have provided confidence intervals for selected variables in the text. Pages 9-10 lines 222 to 229

Comment: Remember to account for the complex nature of this sample.

Response: This comment was not clear.

Comment: c. Table 3 for the models. See the above comments

Response: We have noted the above comments and adjusted for models. Continuous age vs categorical age had no effect on model fit and the coefficients and hence maintained categorical age for that matter. Pages 12-13

Comment: 7. Discussion: why no limitation discussion here?

Response: Study limitations have been added. Page 15 Lines 321-326

Reviewer 2

Comment: The manuscript is technically sound, however to understand the results and the conclusions of the study it will be useful if the authors provide data on HepB policy to prevent HBV in Uganda in general, and in the district in particular:

a) Coverage of hepatitis B virus vaccine

b) Coverage of prevention of mother-to-child transmission of hepatitis virus

c) Blood donations screened in a quality-assured manner

d) Infections administered with safety-engineered devices

Hepatitis protection among HCP is not PEP alone, and a number of strategies include immunization against HepB, universal precautions, control measures, education, and reporting and follow-up of exposure.

Programmes to support Infection, Prevention and Control are particularly important in low-income countries, where health care delivery and medical hygiene standards may be improved. Infection, Prevention and Control Programmes is the key to fight against infectious diseases in general, and against HepB in particular, and PEP is one of the components of the strategy.

Reviewer: Thank you. We have provided data on the hepatitis B policy in Uganda in the background. Page 4 lines 104 to 131. Coverage of hepatitis B vaccination especially among healthcare providers in Wakiso has also been provided. Page 3 lines 97 to 103. Information on coverage of prevention of mother-to-child transmission of hepatitis virus is limited. However, we have acknowledged the relationship between prevention of mother-to-child transmission and risk of hepatitis B infection among the HCPs. Page 3 lines 95-98

Information on whether blood donations are screened in a quality assured manner is provided on Page 4 lines 109-111. Other measures related to protection from hepatitis B have been elaborated on Page 4 lines 108-113

Comment: Finally, are the healthcare facilities where interviews have occurred, did serologic tests for HpB, and HBIG and hepatitis B vaccine are available?

The availability of these data is useful to understand better in which context this study was conducted, and the reasons for the low prevalence of awareness of HepB PEP among HCP in Wakiso district, Central Uganda.

Response: We did not conduct serologic tests for HepB, and HBIG. However, the vaccination status of the respondents has been reported in our earlier study and results summarized on Page 4 lines 102-104

Comment: Is the manuscript technically sound: Yes, although the manuscript would benefit from an English review and editing services to improve the overall flow?

Response: Thank you. We have sought services of an English reviewer and editor

Comment: In the introduction, it would be important to provide data on the approximate mortality from HBV in Uganda in order to give the reader a better perspective of the burden of the disease and to further justify this study.

Response: Thank you. Data on the approximate HBV-related deaths has been added. Page 3 line 80

Comment: The authors should consider providing information on the availability of HBIG and Hepatitis Vaccination at health facilities in Uganda; limited access to PEP may also contribute to the little knowledge.

Response: Information on the availability of the vaccine has been provided on Page 10 Lines 231-232. Information on awareness of HBV has also been provided in our earlier publications, and in a summarized version on Page 4 Lines 104-105

 Comment: The authors should provide a stronger justification for this study. In the introduction, the authors state "These findings can be used to inform policy and practice related to the prevention of HBV infection among HCPs." There are other reasons why it would be important to study the knowledge on HBV prophylaxis and it would be helpful for the authors to elaborate that in this paper to further convince the reader of the importance of this study.

Response: The justification has been improved by elaborating further the importance of the study towards the prevention of HBV infection. Page 137-140

Comment: Methods: The authors note that "Variables with p values less than 0.25 in the bivariable models and those with literature backup evidence were added into the multivariable model while adjusting for age and sex". What informed the choice of p value of 0.25 as a threshold for including variables in the multivariable analysis?

Response: It’s a threshold that has been suggested in literature. But the rationale is really to allow flexible for variables which would not ordinally be significant due to sample size issues and yet would be important predictors in adjusted models.

Comment: From the power and sample size calculation, the authors calculated a desired sample size of 325. However, data was collected from 306 research participants. The authors have not discussed the potential impact of data collection from a smaller population size than initially calculated.

Response: Thanks for the concern but with a response rate of over 94.1%, we believe this wouldn’t substantially affect the proportions and effect estimates.

Comment: The authors should be clearer on the definition of the dependable outcome. They have provided a brief definition of what "knowledge on HBV PEP". This definition however needs to be elaborated.

Response: Dependent variable is well described on page 6 under study variables. We have no variable called Knowledgeable on HBV PREP but have provided several variables which assess different aspects of knowledge and these variables are described under study variables and details are in table 2. Our dependent variable was awareness of PEP options

Comment: Discussion: Should be expanded to include a discussion on why the various social and demographic factors were selected as independent variables, and the significance of the relationship, significant or not, to the dependent variable.

Response: The significant factors have been discussed. Pages 14-15 Lines 289-314

Comment: Recommendations: The authors state that "Our findings suggest the need to use screening and vaccination

opportunities to sensitize HCPs on the PEP options for HBV infection.” There could be other recommendations from this study. The primary recommendation should be to educate health workers on the need and on the availability of HBV PEP, given that 90% of them are not aware. The recommendations could be strengthened to align with the findings from the study.

Response: The recommendations have been strengthened. Page 16 Lines 328-332

---

## [Decision Letter · Decision Letter 1]

20 Apr 2022

PONE-D-21-16912R1Awareness of Hepatitis B Post-Exposure Prophylaxis among Healthcare providers in Wakiso district, Central Uganda.PLOS ONE

Dear Dr. Ssekamatte,

Thank you for submitting your manuscript to PLOS ONE. After careful consideration, we feel that it has merit but does not fully meet PLOS ONE’s publication criteria as it currently stands. Therefore, we invite you to submit a revised version of the manuscript that addresses the points raised during the review process.

We look forward to receiving your revised manuscript.

Kind regards,

Orvalho Augusto, MD, MPH

Academic Editor

PLOS ONE

Journal Requirements:

Reviewers' comments:

Reviewer's Responses to Questions

**Comments to the Author**

1. If the authors have adequately addressed your comments raised in a previous round of review and you feel that this manuscript is now acceptable for publication, you may indicate that here to bypass the “Comments to the Author” section, enter your conflict of interest statement in the “Confidential to Editor” section, and submit your "Accept" recommendation.

Reviewer #1: All comments have been addressed

2. Is the manuscript technically sound, and do the data support the conclusions?

Reviewer #1: Yes

3. Has the statistical analysis been performed appropriately and rigorously? 

Reviewer #1: Yes

4. Have the authors made all data underlying the findings in their manuscript fully available?

Reviewer #1: Yes

5. Is the manuscript presented in an intelligible fashion and written in standard English?

Reviewer #1: Yes

6. Review Comments to the Author

Reviewer #1: Lines 70-72 – Change to “Chronic HBV infection remains one of the most serious of viral hepatitis, and is often associated with hepatocellular necrosis, inflammation, being cirrhosis and hepatocellular carcinoma the major complications”.

Lines 92-96 – Injection practices worldwide and especially in LMICs include multiple, available unsafe practices. Unsafe practices but are not limited to prevalent and high-risk practices, include: a) Reuses of injection equipment to administer injections to more than one person; b) accidental needle stick injuries in health-care workers; c) overuse of injection to health conditions where oral formulations are available; d) unsafe sharps waste management. Prevention of HBV infection in health-care setting include hand hygiene, safe handling and disposal of sharps and waste, safe cleaning of equipment, testing of donated blood, improved access to safe blood and training the health personnel. This manuscript is a opportunity to pass to the healthcare provides a clear message, regarding the awareness of hepatitis B among healthcare providers.

Lines 100-101 – Please correct to “…among HCPs (14,24)”

Line 106 – Please correct to “…post-exposure prophylaxis (PEP), resulting…”.

Lines 116-118 – Please correct to “HBV PEP include prevention of perinatal and early childhood HBV infection, persons who inject drugs, men who have sex with men, sex workers and healthcare providers.”

Line 133 – Please correct to “…seek PEP (35.36%).”

Line 135 – Please correct to “…HBV PEP among HCPs…”

Line 145 – Please correct to “Healthcare facilities (HCFs)…”

Line 146 – Please correct to “…which is designated as HC I, to HC II, III, IV, …”

Line 158 – Please correct to “…adequate knowledge on HBV PEP of 12.1% (40)…”

Line 176 – Please correct to “…management of HBV infection.”

Line 180 – Please correct to “…was awareness of HBV PEP options.”

Line 182 – Please correct to “…HBIG or HBV vaccine or both”

Line 200 – Please correct to “…(awareness of PEP options for HBV)…”

Line 202 – Please correct to “…of PEP options for HBV on…”

Line 233 – Please correct to table 2.

Lines 237,238,241,242,244,245,250,252,255 and tables 3 and 4 – Please switch Hepatitis B to HBV infection

Line 245 – Please correct to table 3

Line 257 – Please correct to table 4

Lines 258 and 259 – Please correct to table 4.

Lines 262, 264-266 – Please switch Hepatitis B to HBV infection.

Line 268 – Please correct to “…exposure to HBV infection (34).”

Line 270 – Please correct to “…HBV PEP.”

Lines 274,277,281,283,285,292,295,296,300,302,304,306-308,313,327,329 and 331 – Please switch hepatitis B to HBV

Lines 296,297,322 and 330 – Please use HCFs instead healthcare facilities.

Lines 309 and 310 – Please use HCPs instead healthcare providers.

7. PLOS authors have the option to publish the peer review history of their article (what does this mean?). If published, this will include your full peer review and any attached files.

Reviewer #1: No

---

## [Author Response · Author response to Decision Letter 1]

27 Apr 2022

Awareness of Hepatitis B Post-Exposure Prophylaxis among Healthcare providers in Wakiso district, Central Uganda

Response to comments/edits

Comment/suggestion Response action

Lines 70-72 – Change to “Chronic HBV infection remains one of the most serious of viral hepatitis, and is often associated with hepatocellular necrosis, inflammation, being cirrhosis and hepatocellular carcinoma the major complications”. Thank you, has been revised accordingly

Lines 92-96 – Injection practices worldwide and especially in LMICs include multiple, available unsafe practices. Unsafe practices but are not limited to prevalent and high-risk practices, include: a) Reuses of injection equipment to administer injections to more than one person; b) accidental needle stick injuries in healthcare workers; c) overuse of injection to health conditions where oral formulations are available; d) unsafe sharps waste management. Prevention of HBV infection in healthcare settings includes hand hygiene, safe handling and disposal of sharps and waste, safe cleaning of equipment, testing of donated blood, improved access to safe blood, and training the health personnel. This manuscript is thus an opportunity to pass to the healthcare providers a clear message regarding awareness of hepatitis B of prevention. Thank you, has been revised accordingly

Lines 100-101 – Please correct to “…among HCPs (14,24)” Thank you, has been revised accordingly

Line 106 – Please correct to “…post-exposure prophylaxis (PEP), resulting…”. Thank you, has been revised accordingly

Lines 116-118 – Please correct to “HBV PEP include prevention of perinatal and early childhood HBV infection, persons who inject drugs, men who have sex with men, sex workers and healthcare providers.” Thank you, has been revised accordingly

Line 133 – Please correct to “…seek PEP (35.36%).” Thank you, has been revised accordingly

Line 135 – Please correct to “…HBV PEP among HCPs…” Thank you, has been revised accordingly

Line 145 – Please correct to “Healthcare facilities (HCFs)…” Thank you, has been revised accordingly

Line 146 – Please correct to “…which is designated as HC I, to HC II, III, IV, …” Thank you, has been revised accordingly

Line 158 – Please correct to “…adequate knowledge on HBV PEP of 12.1% (40)…” Thank you, has been revised accordingly

Line 176 – Please correct to “…management of HBV infection.” Thank you, has been revised accordingly

Line 180 – Please correct to “…was awareness of HBV PEP options.” Thank you, has been revised accordingly

Line 182 – Please correct to “…HBIG or HBV vaccine or both” Thank you, has been revised accordingly

Line 200 – Please correct to “…(awareness of PEP options for HBV)…” Thank you, has been revised accordingly

Line 202 – Please correct to “…of PEP options for HBV on…” Thank you, has been revised accordingly

Line 233 – Please correct to table 2. Thank you, has been revised accordingly

Lines 237,238,241,242,244,245,250,252,255 and tables 3 and 4 – Please switch Hepatitis B to HBV infection Thank you, has been revised accordingly

Line 245 – Please correct to table 3 Thank you, has been revised accordingly

Line 257 – Please correct to table 4 Thank you, has been revised accordingly

Lines 258 and 259 – Please correct to table 4. Thank you, has been revised accordingly

Lines 262, 264-266 – Please switch Hepatitis B to HBV infection. Thank you, has been revised accordingly

Line 268 – Please correct to “…exposure to HBV infection (34).” Thank you, has been revised accordingly

Line 270 – Please correct to “…HBV PEP.” Thank you, has been revised accordingly

Lines 274,277,281,283,285,292,295,296,300,302,304,306-308,313,327,329 and 331 – Please switch hepatitis B to HBV Thank you, has been revised accordingly

Lines 296,297,322 and 330 – Please use HCFs instead healthcare facilities. Thank you, has been revised accordingly

Lines 309 and 310 – Please use HCPs instead healthcare providers. Thank you, has been revised accordingly

---

## [Decision Letter · Decision Letter 2]

10 May 2022

PONE-D-21-16912R2Awareness of Hepatitis B Post-Exposure Prophylaxis among Healthcare providers in Wakiso district, Central Uganda.PLOS ONE

Dear Dr. Ssekamatte,

Thank you for submitting your manuscript to PLOS ONE. After careful consideration, we feel that it has merit but does not fully meet PLOS ONE’s publication criteria as it currently stands. Therefore, we invite you to submit a revised version of the manuscript that addresses the points raised during the review process.

We look forward to receiving your revised manuscript.

Kind regards,

Orvalho Augusto, MD, MPH

Academic Editor

PLOS ONE

Journal Requirements:

Additional Editor Comments (if provided):

There are numerous minor typos and inconsistencies to fix as the reviewer points out below.

Reviewers' comments:

Reviewer's Responses to Questions

**Comments to the Author**

1. If the authors have adequately addressed your comments raised in a previous round of review and you feel that this manuscript is now acceptable for publication, you may indicate that here to bypass the “Comments to the Author” section, enter your conflict of interest statement in the “Confidential to Editor” section, and submit your "Accept" recommendation.

Reviewer #1: (No Response)

2. Is the manuscript technically sound, and do the data support the conclusions?

Reviewer #1: Yes

3. Has the statistical analysis been performed appropriately and rigorously? 

Reviewer #1: Yes

4. Have the authors made all data underlying the findings in their manuscript fully available?

Reviewer #1: Yes

5. Is the manuscript presented in an intelligible fashion and written in standard English?

Reviewer #1: Yes

6. Review Comments to the Author

Reviewer #1: Remain some minor additional comments

Considering that healthcare providers (HCPs) is the same that health care workers (line 135), please uniform to HCPs

When a full name (e.g. continuous medical education – line 144) is used less than five times in the text there is no sense to use the acronym (e.g. CME)

Please uniform to hepatitis B PEP, and do not write the first letter of hepatitis with capital letter

Line 41… risk of hepatitis B virus…

Line 43… awareness of hepatitis B PEP…

Line 47… selected from healthcare facilities (HCPs)…

Line 77… burden of chronic HBV, …

Line 79… in the mortality due to human immunodeficiency virus (HIV)…

Line 81… Hepatitis B infection is…

Line 84… Hepatitis B infection accounts…

Line 89… Healthcare providers in SSA are…

Line 90… Healthcare providers have an up…

Line 93… especially in lower middle – income economies…

Table 1 In addition to services offered at healthcare center IV…

… such as psychiatry, ear, nose and threat, ophthalmology…

General hospital

Regional referral hospital

National referral hospital

Line 112… precautions and PEP, resulting…

Line 120-121… Post-exposure prophylaxis is effective in the prevention

Line 122… Hepatitis B PEP…

Line 131… on PEP for hepatitis B, C and HIV �37

Line 139… awareness of hepatitis B PEP

Line 141… hepatitis B PEP

Line 144… continuous medical education sessions

Lines 150-151… (10 hospitals, 15 health centres (HCs)… Healthcare facilities in Uganda…

Line 163… knowledge on hepatitis B PEP of 12,1% �43�,…

Line 171… private for profit, private not for profit or public…

Line 185… hepatitis B PEP options…

Line 191… Healthcare facilities was considered…

Line 193… Healthcare providers were classified…

Line 205… (awareness of PEP options for hepatitis B)…

Line 206-207… of PEP for hepatitis B

Line 212… 95% confidence intervals are reported…

Line 224… Research and Ethics Committee. Administration…

Line 239 (Table 2) when acronyms are used in Tables (e.g. HCP and HCF), please use the full

name as foot note at the end of the table or alternatively use the full name

Line 242… PEP for hepatitis B infection,…

Line 246… PEP for hepatitis B.

Line 248… hepatitis B infection.

Line 250 (Table 3) It’s recommended to not use acronyms in tables (e.g. HBV, PEP, HBIG),

however if it’s used please write the full name as foot note at the end of the

table

Line 257… PEP options for hepatitis B (AOR = 0.11, …)

Lines 262-263 (Table 4) The same recommendations on the use of acronyms as for tables 1, 2

and 3

Line 266… the awareness of hepatitis B PEP among…

Line 268… awareness of hepatitis B PEP among…

Line 269… that hepatitis B had PEP.

Line 270… of hepatitis B PEP may result…

Line 274… sensitive about hepatitis B PEP.

Line 279… options for hepatitis B �43

Line 288… different hepatitis B PEP options…

Line 292… the hepatitis B PEP options…

Line 299 Healthcare providers in urban HCPs were more likely to be aware of hepatitis B PEP…

Lines 300, 304, 305, 307, 309, 310, and 315… hepatitis B PEP

Line 322… of hepatitis B PEP…

Line 329… hepatitis B PEP

Line 331… hepatitis B. Healthcare providers awareness of hepatitis B PEP…

Line 333, 334 and 335… hepatitis B PEP…

7. PLOS authors have the option to publish the peer review history of their article (what does this mean?). If published, this will include your full peer review and any attached files.

Reviewer #1: No

---

## [Author Response · Author response to Decision Letter 2]

3 Jun 2022

Response to Reviewers

Dear Editor, 

We would like to appreciate the critical review and comments provided by the reviewers. We have carefully addressed each comment as explained in the table below. We feel the quality of the manuscript has greatly improved. The responses have been attached

---

## [Editor Report · Decision Letter 3]

7 Jun 2022

Awareness of Hepatitis B Post-Exposure Prophylaxis among Healthcare providers in Wakiso district, Central Uganda.

PONE-D-21-16912R3

Dear Dr. Ssekamatte,

We’re pleased to inform you that your manuscript has been judged scientifically suitable for publication and will be formally accepted for publication once it meets all outstanding technical requirements.

Kind regards,

Orvalho Augusto, MD, MPH

Academic Editor

PLOS ONE
---

## [Editor Report · Acceptance letter]

13 Jun 2022

PONE-D-21-16912R3 

Awareness of Hepatitis B Post-Exposure Prophylaxis among Healthcare providers in Wakiso district, Central Uganda. 

Dear Dr. Ssekamatte:

I'm pleased to inform you that your manuscript has been deemed suitable for publication in PLOS ONE. Congratulations! Your manuscript is now with our production department. 

Kind regards, 

on behalf of

Dr. Orvalho Augusto 

Academic Editor

PLOS ONE